# Life History and Host Preference of *Trichopria drosophilae* from Southern China, One of the Effective Pupal Parasitoids on the *Drosophila* Species

**DOI:** 10.3390/insects11020103

**Published:** 2020-02-04

**Authors:** Chuandong Yi, Pumo Cai, Jia Lin, Xuxiang Liu, Guofu Ao, Qiwen Zhang, Huimin Xia, Jianquan Yang, Qinge Ji

**Affiliations:** 1Institute of Beneficial Insects, Plant Protection College, Fujian Agriculture and Forestry University, Fuzhou 350002, China; chuandyi@163.com (C.Y.); caipumo@163.com (P.C.); Lin14787861578@163.com (J.L.); liuxx1003@163.com (X.L.); agf1025@163.com (G.A.); Wenww1966@163.com (Q.Z.); 15980626028@163.com (H.X.); 2State Key Laboratory of Ecological Pest Control for Fujian and Taiwan Crops, Fuzhou 350002, China; 3Key Lab of Biopesticide and Chemical Biology, Ministry of Education, Fuzhou 350002, China; 4Department of Horticulture, College of Tea and Food Science, Wuyi University, Wuyishan 354300, China; 5College of Agriculture, Anshun University, Anshun 561000, China

**Keywords:** spotted wing drosophila, biological control, *Trichopria drosophilae*, life table fertility, parasitic preference

## Abstract

This study aims to evaluate several life-history traits of a *T. drosophilae* population from southern China and its parasitic preference of three *Drosophila* species. For mated *T. drosophilae* females, the mean oviposition and parasitization period were 27.20 and 37.80 d, respectively. The daily mean parasitization rate was 59.24% per female and the lifetime number of emerged progeny was 134.30 per female. *Trichopria drosophilae* females survived 37.90 and 71.61 d under host-provided and host-deprived conditions, respectively. To assess the potential for unmated reproduction in *T. drosophilae*, the mean oviposition and parasitization period of unmated females was 22.90 and 47.70 d, respectively. They had a daily mean parasitization rate of 64.68%, produced a total of 114.80 offspring over their lifetime, and survived 52 d. Moreover, *T. drosophilae* showed a preference towards *D. suzukii* based on the total number of emerged offspring under a choice test. Our findings indicate that *T. drosophilae* from southern China appears to be suitable for the control of *D. suzukii* in invaded areas, due to its reproductive potential.

## 1. Introduction

*Drosophila suzukii* (Matsumura) (Diptera: Drosophilidae), the spotted wing drosophila, is a polyphagous pest of soft and thin-skinned fruit crops, with negative economic effects in its native continent of Asia [1]. This species has expanded its geographical range dramatically across America and Europe [1,2,3,4], where it has caused significant economic losses to fruit industries in the newly invaded countries. Unlike most other *Drosophila* flies, which attack decaying or rotting fruits and are generally not regarded as pests, *D. suzukii* females possess a prominent serrated ovipositor that allows it to lay eggs inside unblemished ripening fruit [5]. Visible physical damage is caused directly by both external piercing and internal larval feeding, with the oviposition wounds allowing pathogens to penetrate, thus rendering contaminated fruit unmarketable [2,6]. This species of *Drosophila* has a short generation time, high productive potential, great ecological adaptability, and wide host range [5,7,8], making effective control of this pest extremely challenging.

Following the invasion of *D. suzukii*, many control efforts have been attempted in order to effectively manage the population; despite this, farmers rely heavily on pesticides such as spinosyns, organophosphates, and pyrethroids for suppressing the pest population [9,10,11]. However, the repeated and extensive use of insecticides may cause a series of secondary problems, such as that from pesticide residues, insect resistance, pest resurgence, and negative effects on pollinators and natural enemies [12,13,14]. Alternative management measures such as preventive and sanitation strategies, mass trapping, sterile insect releases and biological control are currently under evaluation [15,16,17,18,19,20]. In this context, developing area-wide integrated pest management programs (IPM) for this important and mobile pest are urgently needed [9]. Suppressing *D. suzukii* populations through parasitism and predation would assist in improving the outcome of other management practices [21]. However, within-farm mass-release of ground predators is not realistic in the short term, although they can be conserved through cultural strategies [22]. In particular, the application of parasitoids in biocontrol programs may be a more promising avenue, and help to reduce pest populations even in reservoir areas surrounding crop fields [16].

Approximately 50 hymenopterous parasitoid species have been documented attacking various drosophilid species worldwide, the majority of which are larval parasitoids in the families Braconidae and Figitidae and pupal parasitoids in the families Diapriidae and Pteromalidae [23,24,25]. In Europe and North America, several recent studies revealed that most local larval parasitoids failed to develop from *D. suzukii*, mainly due to its strong immune defense under both laboratory conditions and in the field, whereas pupal parasitoids were not found to experience host immune-based defense [9,26,27,28,29]. Recently, an important endoparasitoid of Drosophilidae, *Trichopria drosophilae* Perkins (Hymenoptera: Diapriidae), appeared to be a preferable candidate as a biological control agent against *D. suzukii* [16,17]. Previous studies have also shown that *T. drosophilae* can be more efficient in suppressing *D. suzukii* in laboratory investigations in comparison to other well-known pupal parasitoids such as *Pachycrepoideus vindemiae* Rondani (Hymenoptera: Pteromalidae) [17,28,30], and have also exhibited some promise for *D. suzukii* management in the Italian fruit production industry [21]. Unlike the generalist parasitoid *P. vindemiae*, the host range of *T. drosophilae* is known to be restricted to the family Drosophilae [23,25], and preferentially attacked *D. suzukii* in two-way choice parasitization assays [17]. This parasitoid species is able to parasitize *D. suzukii* at a wide temperature range of 15 to 30 °C. This illustrates that it could be released early in the fruit growing season to deal with the first *D. suzukii* generation, when pest populations are still low after the overwinter bottleneck [11,31]. 

*Trichopria drosophilae* seems to be cosmopolitan with a wide distribution, and has been reported to attack pupae of *D. suzukii* in Europe [28], North America [16,17], and Asia [32,33,34] The effectiveness against *D. suzukii* may vary between different geographical populations of parasitoids [33]. For example, based on offspring survival, development, and reproduction, a Californian population of *P. vindemiae* was more heat-tolerant but less cold-tolerant than an Oregon population when provided with *D. suzukii* pupae as hosts [35]. Furthermore, the larval parasitoids *Leptopilina heterotoma* also differed, based on their geographical origin, in their ability to parasitize *D. suzukii* larvae: an Italian population of this parasitoid could successfully parasitize the pest [28], whereas a French population was unable to do so [26]. Wang [16] compared the age-specific lifetime fertility of *T. drosophilae* populations from invaded and native regions of *D. suzukii*; they found that the Californian colony of *T. drosophilae* appeared to perform better than the South Korean colony based on their reproductive potential. Although *T. drosophilae* has been recorded as effective against *D. suzukii* in different areas under a series of laboratory tests [16,17,21,28,29,33,35,36], it is still necessary to understand the basic biological attributes of this parasitoid originating from other locations because parasitoids from different geographical locations can have different performance on *D. suzukii* [33]. Additionally, this aids in screening for the most suitable populations for effective rearing and augmentative field release against *D. suzukii*. 

Recently, we surveyed parasitoid species attacking *D. suzukii* in several waxberry (*Myrica rubra*) orchards located within Fujian Province, China, using traps baited with *D. suzukii* pupae and larvae. We found that *T. drosophilae* was dominant among the collected parasitoid samples, followed by several unknown *Asobara* spp. (unpublished data). The indigenous colony of *T. drosophilae* from Fujian berry production areas was successfully established using *D. suzukii* pupae as hosts in our laboratory. The objective of this study is to evaluate the biological traits of promising southern Chinese populations of *T. drosophilae* by assessing its effectiveness in controlling *D. suzukii*. Our findings, we hope, will contribute to enhancing biological control strategies against *D. suzukii* by the use of resident hymenopteran parasitoids from the pest’s native areas. 

## 2. Materials and Methods

### 2.1. Insects

Laboratory colonies of *D. suzukii*, *Drosophila melanogaster* Meigen, *Drosophila immigrans* Sturtevant (Diptera: Drosophilidae), and *T. drosophilae* were established from field collections that took place during 2016 from waxberry orchards in Zhangzhou City, Fujian Province, China, and maintained under controlled conditions (25 ± 1 °C, 70 ± 5%RH, 16:8 h (L:D)). *D**rosophila suzukii* populations were collected from infested waxberries; *D. melanogaster* and *D. immigrans* were collected from rotten berries. The parasitoid populations were collected from sentinel traps baited with *D. suzukii* pupae.

The four species were reared according to rearing methods previously described by Wang [16,17]. Briefly, adult flies were maintained in 1800-mL glass containers and supplemented with a standard cornmeal-based artificial diet (refs. [16,17]; brewery yeast 20 g, cornmeal 50 g, sucrose 40 g, agar 10 g, 36% *v*/*v* acetic acid 3 mL, 95% *v*/*v* ethanol 6.7 mL, and potassium sorbate 1 g). Apple pulps were deposited in the middle of containers for the oviposition of adult flies. Adult parasitoids were held in “Hawaii-type” cages (30 × 30 × 30 cm) [37] and supplied with a 50% honey-water solution in a vial as nutrition. After the *D. suzukii* progeny had developed into two-day-old pupae (5–7 days after oviposition), they were selected randomly and seeded on wet Petri dishes (100 mm), and then exposed to adult parasitoids for two days. Subsequently, the parasitoid-exposed dishes were transferred to new cages and maintained for parasitoid emergence. All experiments were also carried out under the rearing conditions for flies and parasitoids.

### 2.2. Parasitoid Reproduction Performance and Longevity

The longevity and age-specific lifetime fertility of *T. drosophilae* females originating from Fujian Province, China, was investigated using two-day-old *D. suzukii* pupae as hosts since it has been suggested that pupae age does not influence the parasitization rate of this parasitoid [16]. Newly emerged parasitoids were paired and isolated in 100-mm Petri dishes, together with a honey-soaked cotton ball deposited inside as food and 30 *D. suzukii* pupae as hosts. To verify the reproduction potential of the unmated parasitoid, virgin females were individually isolated under the same conditions. In total, 40 pairs of mated parasitoids and 20 virgin females were used. Of the 40 pairs, a subset of 20 pairs was host-deprived, while the remaining 20 pairs were constantly supplied with 30 host pupae. In the cases where the male parasitoid died before the female, a newly emerged male was substituted. Every 24 h, host pupae exposed to the parasitoids (not deprived treatment) were renewed until all female parasitoids died. On each host replacement, the parasitoids were transferred to a new dish including fresh hosts and new cotton wick soaked with honey solution. The Petri dishes of the host-deprived parasitoids were replaced every day to ensure good conditions and to avoid the cotton from turning moldy. All exposed hosts were held in their original dishes and kept in an incubation chamber under the culture conditions described above until either adult flies or parasitoids emerged. One additional set of ten control dishes containing two-day-old pupae were deposited daily alongside the parasitization dishes, in order to test the *D. suzukii* daily emergence rate in the absence of parasitism. 

All dishes were inspected daily and the longevity of females, number and sex of parasitoid offspring, survival rate, and number of parasitized hosts were recorded. The parasitization rate was estimated as the total number of host pupae used by parasitoids (containing those successfully emerging and failing to emerge) divided by the number of host pupae available. After adult emergence had ceased, all unemerged pupae were dissected and the parasitoid offspring that failed to emerge were documented. In cases where it was difficult to distinguish unparasitized from parasitized pupae (e.g., when the parasitoid offspring died in the early developmental stages), their parasitization status was considered as “unknown”. The sex ratio, expressed as the proportion of females, was also estimated by determining the sex of both emerged parasitoids and unemerged pupae (when discernible). The mean number of offspring produced each day was calculated in terms of the total number of offspring produced during each one-day exposure. Oviposition duration was estimated and analyzed according to the methods described by Núňez-Campero et al. [38]. Parasitized host pupae were readily recognizable prior to the emergence of parasitoid offspring due to a darkening of the pupae from the parasitoids inside. If the fly emergence percentage was significantly lower in the presence rather than absence of parasitoids, it was considered that parasitization had occurred. The parasitization period was defined as the number of days for which the host emergence was significantly lower in the treatment group than in the control group.

Based on the collected data, standard life table fertility parameters were calculated, namely, the intrinsic rate of natural increase (*r*), net reproductive rate (*R_o_*), mean generation time (*T*) and doubling time (*DT*). The value of *r* was calculated as $\sum e^{-rx}l_{x}m_{x}=1$, where *x* is expressed as female age in days, *lx* is expressed as the age-specific survival rate, and *mx* is the number of daughters produced per female alive at age *x* [39]. *R_o_* was estimated as $R_{o}=\sum l_{x}m_{x}$, *T* in days as *T* = ln*R_o_*/*r*, and TD in days was given by *DT* = ln(2)/*r* [12,16].

### 2.3. Host Species Preference and Parasitic Efficiency

A choice test was performed to test whether *T. drosophilae* had a preference for a particular species of *Drosophila* pupae, and the effect of the host species on the parasitoid’s offspring fitness (emergence rate and sex ratio). Newly emerged parasitoids were paired and introduced in the bioassay cage (30 × 30 × 30 cm) provisioned with a honey-soaked streak as nutrition for adult parasitoids. For each replicate, 30 pupae from each of three *Drosophila* species (*D. suzukii*, *D. melanogaster*, *D. immigrans*) were deposited on wet filter paper and simultaneously exposed to a pair of parasitoids for 24 h within the same bioassay cage. The pupae of each *Drosophila* species were both two-day-old and randomly arranged in a regular triangle shape with 10 cm sides on the filter paper. To eliminate the effects caused by position bias, the tested *Drosophila* species were altered in their positions in different replicates. Following exposure, the host pupae from different species were maintained separately on wet filter paper within clean Petri dishes and held for fly or adult parasitoid emergence. The number and sex of emerged parasitoids were documented three times per day (8:00, 16:00, 24:00 h). A total of 20 replicates were performed.

A no-choice test was conducted to determine the parasitic efficiency of *T. drosophilae* when two-day-old *D. suzukii*, *D. melanogaster*, and *D. immigrans* pupae were respectively offered as hosts. For each replicate, 30 pupae of each host species were deposited on a wet tissue paper inside a 100-mm Petri dish and exposed to a pair of parasitoids for 24 h, with honey-soaked cotton as food for the adult parasitoids. Five-day-old parasitoids were used. After the 24 h exposure period, each *Drosophila* species was held separately in Petri dishes until the emergence of flies or adult parasitoids. The number and sex of emerged parasitoids were recorded three times per day. Twenty-eight replicates were performed for each *Drosophila* species.

### 2.4. Data Analysis

Trends in the parasitism rate, offspring emergence rate, and sex ratio throughout the parasitoid’s lifespan were analyzed through linear regression. In this case, only vials in which either adult emergence occurred or whereby the sex of dead parasitoids within the host pupae was discernible were considered, while the other vials were excluded from the analysis. Longevity was analyzed using the log-rank test. The effect of host species on the number of offspring produced or offspring sex ratio was analyzed using an analysis of variance (ANOVA). All analyses were conducted using SPSS v.17.0 (SPSS Inc., Chicago, IL, USA) and WPS Office 2017 (Kingsoft Co. Ltd., Beijing, China).

## 3. Results

### 3.1. Parasitoid Reproductive Potential and Longevity

Regardless of whether *T. drosophilae* females had mated, they began oviposition within 24 h of emergence, without an obvious pre-oviposition period. The average oviposition period recorded for *T. drosophilae* mated females was 27.20 ± 1.59 d (range: 19–33 d), which was significantly lower than the mean parasitization period of 37.80 ± 3.64 d (range: 20–48 d; *F* = 2.67, df = 11.06, *p* < 0.05). *T. drosophilae* unmated females had a mean oviposition period of 22.90 ± 2.18 d (range: 20–26 d), also significantly lower than the mean parasitization period of 47.70 ± 5.48 d (range: 33–52 d; *F* = 6.28, df = 18, *p* < 0.05; Figure 1). 

The daily average parasitization rate documented for *T. drosophilae* females mating with males was 59.24% ± 22.50% per female; the average rate was >20% from the 1st to the 25th day and reached its highest value on the 4th day. With respect to *T. drosophilae* females which did not mate, the daily mean parasitization rate was 64.68% ± 15.60% per female; the mean rate was > 20% from 1st to the 20th day and reached its highest value on the 6th day. As parasitoid females aged, the parasitization rate recorded for both with or without copulation showed a decreasing tendency over the lifespan of the parasitoid (male and female mating: *F* = 23.19, df = 1, *p* < 0.05; unmated: *F* = 84.20, df = 1, *p* < 0.05; Figure 2).

The lifetime number of emerged progeny recorded for *T. drosophilae* parents that mated with males was 134.30 ± 7.14 (range: 86–178) per female. Of these progeny, 78.50 ± 11.33 were female (range: 11–125) and 55.80 ± 10.76 were males (range: 21–132). The number of both female and male progeny produced by *T. drosophilae* decreased with increasing female age and stopped after 31 and 33 days, respectively (female: *F* = 104.10, df = 1, *p* < 0.05; male: *F* = 6.63, df = 1, *p* < 0.05, Figure 3A). The percentage of female progeny was 58.58% ± 7.45%, which reduced over the lifespan of the female (*F* = 341.22, df = 1, *p* < 0.05; Figure 4). For *T. drosophilae* parents without copulation, the total number of parasitoid offspring that emerged during a lifetime was 114.80 ± 6.23 per female. They only produced male progeny and the production of progeny was significantly decreased over the lifespan of female adults (*F* = 167.34, df = 1, *p* < 0.05; Figure 3B). 

*Trichopria drosophilae* females which mated with males lived significantly longer when host-deprived. Under host-provided conditions, *T. drosophilae* females without copulation lived significantly longer than those which mated with males (χ2 = 10.89, df = 2, *p* < 0.05; Figure 5). The longevity of mated and unmated adult females under host-provided conditions was 37.90 ± 4.45 d (N = 20, range: 20–53) and 52.00 ± 2.23 d (N = 20, range: 33–56), respectively, whereas adult females which mated with males had a greater lifespan of 71.61 ± 4.39 d (N = 20, range: 28–111) under host-deprived conditions. 

For the group of mated females, parameters to evaluate population growth were obtained from the collected data. The net reproductive rate (*R*0) was 52.6; the intrinsic rate of natural increase (*r*) and the finite rate of increase (λ) were 0.16 and 1.18, respectively. Mean generation time (*T*) and population doubling time (*DT*) were 26.29 and 4.23 d, respectively. 

### 3.2. Host Species Preference and Parasitic Efficiency

In the choice test, *T. drosophilae* showed a parasitization preference among three different host species based on the total number of offspring produced. Significantly more parasitoid offspring emerged from *D. suzukii*, followed by *D. immigrans* and *D. melanogaster* (*F* = 28.56, df = 2, *p* < 0.05; Figure 6A). The percentages of female parasitoid offspring from *D. suzukii* and *D. immigrans* were significantly higher than from *D. melanogaster*, and all were over 60% (*F* = 2.69, df = 2, *p* < 0.05; Figure 6A). In the no-choice test, the number of emerged parasitoid offspring derived from *D. melanogaster* pupae was significantly lower than for *D. suzukii* and *D. immigrans* (*F* = 17.60, df = 2, *p* < 0.05; Figure 6B). No difference was observed in the percentage of female parasitoid offspring from *D. suzukii*, *D. immigrans*, or *D. melanogaster*, and all ratios were over 65% (*F* = 1.56, df = 2, *p* > 0.05; Figure 6B).

## 4. Discussion

Biological control strategies, especially those involving parasitoids, remain unutilized in the framework of *D. suzukii* management [21]. Thus far, two different methods have been considered. The first option, which uses simple and reliable tactics, is based on the strengthening of extant parasitoid populations in the newly invaded regions [38]. In fact, on the basis of the Enemy Release Hypothesis [39], these indigenous populations may gradually adapt to new local ecological conditions, establishing and enhancing new relations with the invaded pest, and will, therefore, aid in its demographic control [26,28,40]. Nevertheless, the adaptation process of resident natural enemies in invaded areas requires a variable amount of time, dependant on the plasticity level of the parasitoid [41]. A very weak and unspecific interaction between *D. suzukii* and local parasitoids has been recorded by numerous studies conducted in several invaded regions [23,25,27,28,42,43,44], explained by a short co-adaption time. The second option, namely “classic biological control”, is based on the introduction and continuous construction of natural enemies originating from *D. suzukii* native regions [32,34]. This option is more desirable than the first since specialized resident parasitoids are often absent in the newly invaded areas [1]. Although biological control programs have benefited from the introduction and application of specialized parasitoids, this measure is limited by strict laws regulating the importation of alien species, including biocontrol agents [42]. 

*Trichopria drosophilae* has been demonstrated as a pupal parasitoid that can readily attack many species belong to the family of Drosophilidae [16]. Numerous studies have proved that this common parasitoid can parasitize and develop in countries native to *D. suzukii*, including South Korea [16,32] and China [33], as well as in areas invaded by the pest, such as Europe (Spain [44], Italy [29], France [26]) and North America (Mexico [23], USA [16,17]). Several authors have indicated that *T. drosophilae* could be utilized as an effective biological agent with a great potential for substantially depressing the *D. suzukii* population in the field, particularly in habitats that are surrounded with host crops [16,21,29]. Based on previous studies, *T. drosophilae* females are more efficient at foraging hosts and possess a greater parasitic ability compared to another cosmopolitan pupal parasitoid, *P. vindemiae* [17].

Parasitoids from different areas can impose different effects on *D. suzukii* [33]. As such, it is necessary to identify and screen efficient biological control agents from different locations before importing and applying exotic parasitoids in regions invaded by the pest. In comparison with the *T. drosophilae* population from California, South Korea, and the central area of China, the *T. drosophilae* tested in this work produced almost two-fold more offspring per female, lived longer under both host-provided and host-deprived conditions, and had different life table fertility parameters (Table 1). The differences between the same parasitoid from different areas could be partly due to the different experimental conditions applied in evaluations and genetic differences between the regional biotype. For example, Wang [16] performed their life history determinations at 23 °C and 65%RH, whereas our evaluations were carried out at 25 °C and 70%RH. Considering that the number of offspring produced by *T. drosophilae* was two-fold greater, we supposed that the population from Fujian province, located in southern China, would have a greater reproductive potential compared to the population from other areas. Asplen [1] even partly attributed the different levels of damage caused by *D. suzukii* in China, in comparison to the European and North American invasions, to the occurrence of more efficient natural enemies (especially specialized parasitoids) in the top-down management of pest populations. Combined with our field observations in waxberry orchards in Fujian province, with a predominant *T. drosophilae* population (unpublished data), *T. drosophilae* originating from Fujian would be a favorable choice with which to suppress *D. suzukii* in countries recently invaded by this pest, although further field experiments are needed.

Fecundity is the maximum potential reproductive yield of parasitoid females through their lifespan and is a key parasitic feature [36]. The *Trichopria drosophilae* population used in our experiments showed a similar daily fecundity pattern to that of the Italian and South Korean *T. drosophilae* populations investigated by Rossi Stacconi [42] and Wang [16]. All populations exhibited a peak at an early stage, followed by a reducing tendency and suspension of any parasitization action halfway through their lifespan [45]. A plausible hypothesis to explain the oviposition pattern may be that the egg maturation rate decreases dramatically, which does not compensate timely for egg-load depletion [45]. A previous study found that *T. drosophilae* females emerged with a relatively high mature egg load, and the number of mature eggs was boosted during the first four days [16], which could explain peak fecundity at day four in our experiment. Parasitoids that possess a high load of available mature eggs may maximize their reproductive output at their early developmental stage [16].

Indeed, Driessen and Hemerik [46] suggested that egg depletion should drive the parasitoid to be more selective in choosing a host species. Recently it was reported that *T. drosophilae* presents a parasitization preference for *D. suzukii*, as also shown in this study, probably because of the difference in dimensions in comparison to *D. melanogaster* [17,28]. The preference for large hosts may lead to an increased infestation of *T. drosophilae* on *D. suzukii* in the long-term and ultimately result in increasing the effect of *D. suzukii* suppression. The larger size of *D. suzukii* pupae, compared to pupae of other tested *Drosophila* species, appears to be a preferential characteristic driving the host selection of *T. drosophilae*. Host species selection behavior has significant consequences for the size-fitness association in generalist parasitoids. In *T. drosophilae*, the size of the host fly species was shown to positively correlate with the size of emerged parasitoid offspring [16,35]. Using a large host appeared to come at no cost in terms of developmental time or progeny survival [16]. This mirrors the flexibility of body growth for *T. drosophilae* and appears to be reasonable for this generalist parasitoid; such flexibility in progeny size, incorporated with a probable fitness advantage of becoming large, should induce the evolution of body size and ultimately lead to favoring a large host species [47]. Nevertheless, host species selection behavior is affected by many other factors. On one hand, the occurrence of plasticity in body growth can permit generalist parasitoids to infest an expanded host range; on the other hand, a preference for large host species can narrow host availability [16]. Moreover, physical or physiological ability may inflict constraints on the minimum dimension of hosts and the maximum dimension of parasitoids [48]. More elaborate research with a range of host species of different sizes is required to evaluate the possible physical or physiological ability of this generalist parasitoid and how or to what extent host species selection influences the fitness of *T. drosophilae*. Because of its behavioral simplicity, *T. drosophilae* could serve as a model in which it is practical to quantify the fitness consequence of body size-dependent host selection, and would help to understand the ecology and evolution of host selection. As shown in previous studies [16,35], this parasitoid can be readily raised on a range of *Drosophila* species, meaning their production in an insectary for the purpose of field mass-release as a biocontrol agent is made much easier; most *Drosophila* species have a worldwide distribution and are not difficult to rear in abundance [35]. Additionally, the population of these three tested *Drosophila* species, which were predominant during our field surveys in waxberry orchards (unpublished data), are also predominant in other areas in China such as Guizhou, Yunnan, and Shaanxi [49,50]. Therefore, we decided to perform the choice and no-choice assays, which were conducted on these dominant *Drosophila* species to verify the preference of *T. drosophilae*.

For the strains of mated *T. drosophilae* females, host-deprived females of this parasitoid lived significantly longer than host-provided females, in-line with a previous study [37]. The prolonged lifespan of host-deprived *T. drosophilae* females may be attributed to egg-resorption [51]. In fact, each ovariole has a restricted egg storage capability at any time, and mature eggs can only be maintained in the lateral oviducts for a short period. Thus, when females were kept without hosts for a period longer than the egg maintenance period, the mature eggs began to return to the body of the parasitoid, with only a partial cost in terms of material and energy expenditure [52], resulting in an extension of the lifespan of the parasitoid. Moreover, *T. drosophilae* showed higher longevity for non-copulating females compared to those mating with males. Mating behavior could impose significant effects on the longevity of parasitoid females and the production of female offspring [53]. Although we did not observe the number of copulation events occurring between parasitoid pairs, it was documented that parasitoid females imprisoned in small environments may tend to accept a higher frequency of copulation [12]. Additionally, the percentage of female offspring decreased with increasing parental age. *Trichopria drosophilae* has a sex determination mechanism in which females develop from fertilized eggs while males develop from unfertilized eggs. In most parasitoids, irrespective of host species, the female to male sex ratio reduces with maternal age, since unfertilized eggs can be produced for several days after the production of fertilized eggs has stopped [54]. Although sex ratio determination is attributed to several factors, the main determining factor for sex ratio shifts towards a male-bias in parasitoids exposed to a high density of hosts for a long duration is the depletion of sperm or spermathecal gland secretions [55,56]. Otherwise, the daily fecundity of *T. drosophilae* showed a decreasing tendency with female aging, indicating that the mature egg load was exhausted as age increased. The lower fecundity at parasitoids’ early stage could be compensated for by either an improvement in oviposition later in life or prolonging the reproductive duration.

## 5. Conclusions

This is a study to test the life history and fly-host preference for *a T. drosophilae* population from Fujian province, southern China. These two aspects of the experimental evidence suggest that *T. drosophilae* is one of the effective pupal parasitoids on the *Drosophila* species. Therefore, based on our research, the *T. drosophilae* strain from southern China appears to be a highly promising candidate for application in *D. suzukii* biological control programs. This is due to their greater longevity and stronger oviposition ability than the species from other areas. However, extensive open field investigations considering all the ecological, environmental, and agronomic factors that may influence parasitoid performance are required to confirm the potential beneficial effect of *T. drosophilae* as a biocontrol agent against *D. suzukii*. Further experiments are required to estimate the comparative performance of this parasitoid from different geographical locations against *D. suzukii*. The aim is to maximize the biological control efficiency of *T. drosophilae*.

## Figures and Tables

**Figure 1 insects-11-00103-f001:**
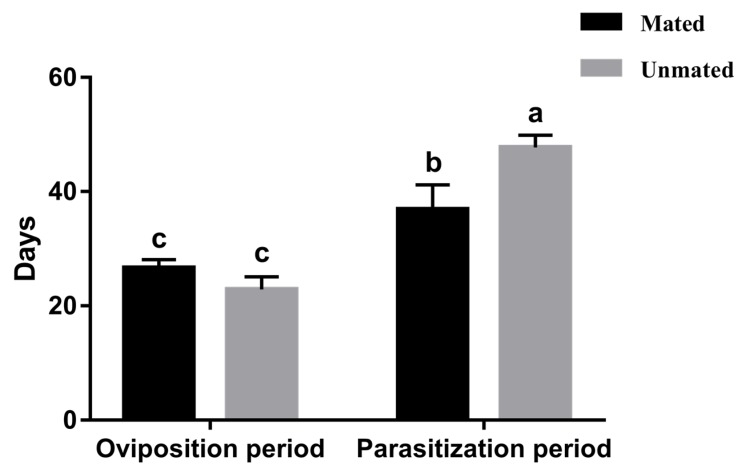
Oviposition and parasitization periods of mated and unmated *T. drosophilae* females. Bars refer to mean ± SE and different letters above the bars indicate significant differences (Tukey’s HSD, *p* < 0.05).

**Figure 2 insects-11-00103-f002:**
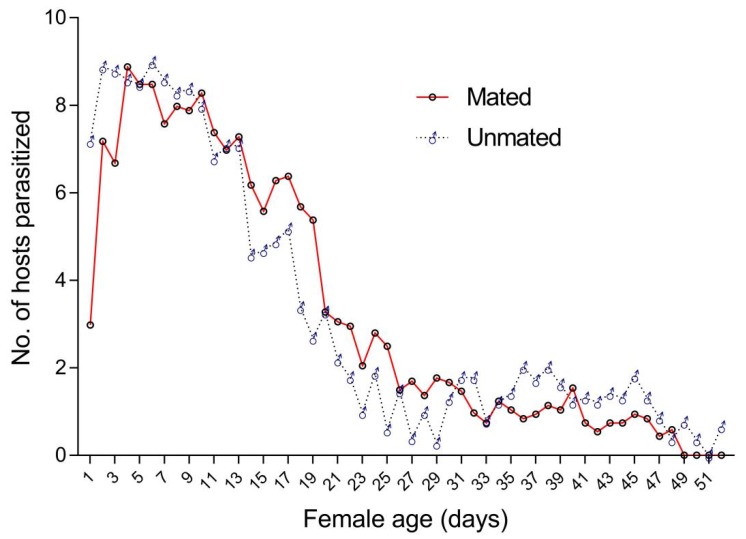
Number of hosts parasitized by mated and unmated *T. drosophilae* females over their life-times.

**Figure 3 insects-11-00103-f003:**
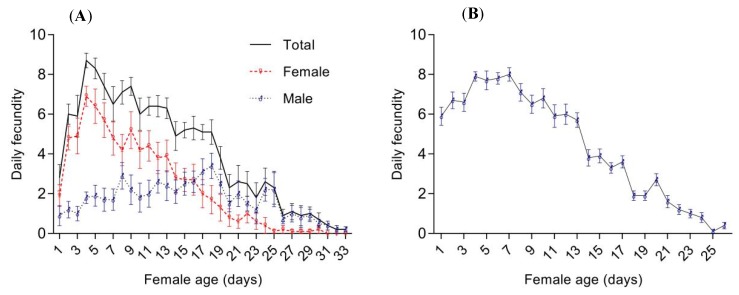
The daily fecundity of *T. drosophilae* females with (**A**) or without (**B**) males. Data are mean ± SE.

**Figure 4 insects-11-00103-f004:**
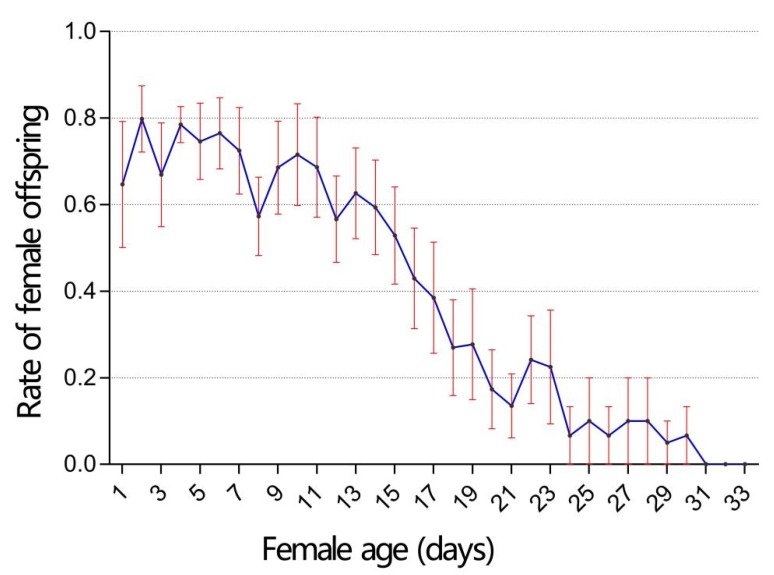
Rate of female offspring of *T. drosophilae* during the lifespan of adult females.

**Figure 5 insects-11-00103-f005:**
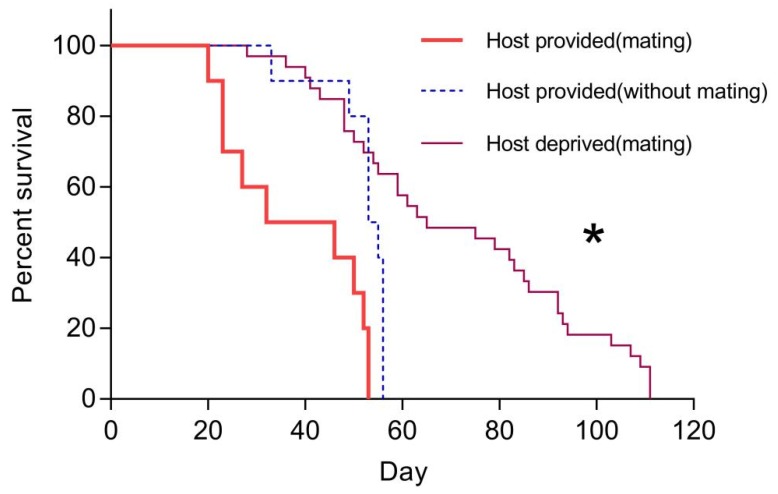
Survival percentages for adult *T. drosophilae* females under host-deprived and host-provided conditions. The asterisk indicates a significant difference between treatments (log-rank test, *p* < 0.05).

**Figure 6 insects-11-00103-f006:**
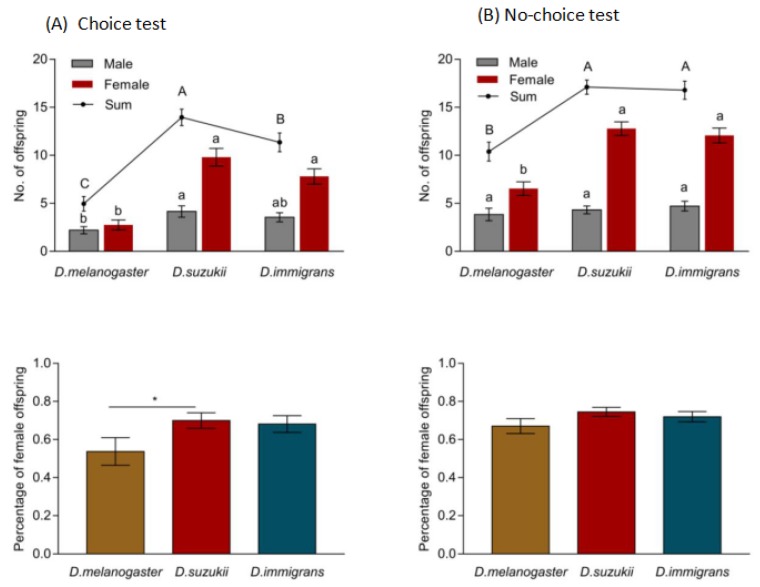
The number and sex ratios of emerged *T. drosophilae* offspring from different host species under choice and no-choice conditions. Different capital or lowercase letters and the asterisk indicate a significant difference between treatments (Tukey’s HSD, *p* < 0.05). Values are mean ± SE.

**Table 1 insects-11-00103-t001:** Main parameters of population growth of *T. drosophilae* from different regional biotypes.

Parameters	Californian [16]	South Korean [16]	Central China [32]	Southern China (This Article)
Rearing temperature	23 °C	23 °C	25 °C	25 °C
Net reproductive rate (*R*_0_)	31.5	22.8	43.75	52.6
Intrinsic rate of natural increase (*r*)	0.124	0.113	0.180	0.164
Generation time (*T*)	27.8	27.7	21.29	26.3
Population doubling time (*DT*)	5.6	6.1	3.91	4.23
Finite rate of increase (λ)	--	--	1.19	1.18
Females longevity	27.5	20.2	22.4	37.90 ± 4.45
Total offspring per female	63.8	52.0	63.45	134.30 ± 7.14

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
