# Peer review of "Life History and Host Preference of Trichopria drosophilae from Southern China, One of the Effective Pupal Parasitoids on the Drosophila Species"

_insects, 2020, doi:10.3390/insects11020103_

Round 1

Reviewer 1 Report

Although the study is well done and the paper well written, my major concern is about the fact that the authors have not used several Trichopria drosophilae populations? Since they mentioned at the end of the introduction: “The objective of this study was to evaluate the biological traits of southern Chinese populations of T. drosophilae...” But they have not used different populations of this parasitoid species? They used a mixture of specimen originated from the same region. They might for example obtained different results on host preference if they were comparing different populations of this parasitoid. Nevertheless, this aspect has been finally discussed from line 293: “Parasitoids from different areas can impose different effects on D. suzukii”. However, to avoid this missing aspect feeling on populations effects in this study to the lectors, I suggest the authors to move at the introduction all this discussion from line 293 to 313 (including the table 1). And then to justify the fact that they will use specimen from the most promising population of Fujian region at Mat & Met section. Also, then, preferably to say at the end of the introduction: “The objective of this study was to evaluate the biological traits of a promising southern Chinese population of T. drosophilae...”.

Therefore, the following sentence at line 392-393 at the conclusion sub-section: “Further experiments are required to estimate the comparative performance of this parasitoid from different geographical locations on D. suzukii.” appears curious to still mentioning it? OR you mean from different geographical location of its host? This has to be deleted or rephrase.

I have also the minor following remarks:

In the abstract, the authors stated that “T. drosophilae showed a preference towards D. suzukii based on the total number of emerged offspring “? I am not fully agreed with that statement since there was no difference between D. suzukii and D. immigrans on the number of offspring under no-choice test (cf figure 6b)... please change it.

In the entire manuscript when the authors start the sentence with a genera name of an insect species they might write it entirely and not in abbreviate even if it has been cited before (e.g; at line 104 at page 3 write “Drosophila suzukii populations...” instead “D. suzukii populations...”.

Reviewer 2 Report

Dear authors,

I find that the introduction of the paper is well elaborated. However I have some doubts about the methodology, You used a log-rank test for the longevity, I suggest you to use a Kaplan Meier survival test, it is more accurate for this kind of analysis.

In the results figure three is not clear enough; I think you can merge figures three and four, since sex ratio (figure 4) is also described indirectly in figure 3. I do not understand either the figure 6.

I would use the sex ratio or female percentage graph and add between brackets the total number of offspring.

The discussion it is okay, but I would modify the conclusion. I think, it is incorrect to use the past to explain a conclusion of a work.
